# Predicting Forest Cover in Distinct Ecosystems: The Potential of Multi-Source Sentinel-1 and -2 Data Fusion

**Kai Heckel [1,2,*], Marcel Urban [2]** **, Patrick Schratz [3,4], Miguel D. Mahecha [5,6] and Christiane Schmullius [2]**

[1] International Max Planck Research School (IMPRS) for Global Biogeochemical Cycles, Max Planck Institute for Biogeochemistry, Hans-Knoell-Str. 10, 07745 Jena, Germany

[2] Department for Earth Observation, Friedrich Schiller University, Grietgasse 6, 07743 Jena, Germany; marcel.urban@uni-jena.de (M.U.); c.schmullius@uni-jena.de (C.S.)

[3] GIScience Group, Friedrich Schiller University, Grietgasse 6, 07743 Jena, Germany; patrick.schratz@uni-jena.de

[4] Department of Statistics, Computational Statistics Group, Ludwig-Maximilian University, 80539 Munich, Germany

[5] Max Planck Institute for Biogeochemistry, Hans-Knoell-Straße 10, 07745 Jena, Germany; mmahecha@bgc-jena.mpg.de

[6] German Centre for Integrative Biodiversity Research (iDiv) Halle-Jena-Leipzig, 04103 Leipzig, Germany

[*] Correspondence: kai.heckel@uni-jena.de; Tel.: +49-3641-9-48975

**Abstract:** The fusion of microwave and optical data sets is expected to provide great potential for the derivation of forest cover around the globe. As Sentinel-1 and Sentinel-2 are now both operating in twin mode, they can provide an unprecedented data source to build dense spatial and temporal high-resolution time series across a variety of wavelengths. This study investigates (i) the ability of the individual sensors and (ii) their joint potential to delineate forest cover for study sites in two highly varied landscapes located in Germany (temperate dense mixed forests) and South Africa (open savanna woody vegetation and forest plantations). We used multi-temporal Sentinel-1 and single time steps of Sentinel-2 data in combination to derive accurate forest/non-forest (FNF) information via machine-learning classifiers. The forest classification accuracies were 90.9% and 93.2% for South Africa and Thuringia, respectively, estimated while using autocorrelation corrected spatial cross-validation (CV) for the fused data set. Sentinel-1 only classifications provided the lowest overall accuracy of 87.5%, while Sentinel-2 based classifications led to higher accuracies of 91.9%. Sentinel-2 short-wave infrared (SWIR) channels, biophysical parameters (Leaf Area Index (LAI), and Fraction of Absorbed Photosynthetically Active Radiation (FAPAR)) and the lower spectrum of the Sentinel-1 synthetic aperture radar (SAR) time series were found to be most distinctive in the detection of forest cover. In contrast to homogenous forests sites, Sentinel-1 time series information improved forest cover predictions in open savanna-like environments with heterogeneous regional features. The presented approach proved to be robust and it displayed the benefit of fusing optical and SAR data at high spatial resolution.

**Keywords:** forest cover; Sentinel-1; Sentinel-2; data fusion; machine-learning; Germany; South Africa; temperate forest; savanna

## 1. Introduction

According to the Food and Agriculture Organization of the United Nations (FAO), approximately one-third of global land area is covered by forests, yet exhibiting a decreasing trend since the 1990's [1].

These estimates include decrease in forest area by deforestation and increase by afforestation. Forests are highly vulnerable ecosystems that are not only habitat to a large number of species and the most widely distributed terrestrial type of vegetation, but also act as a key control in the global carbon and water cycle, hence shaping land-atmosphere feedbacks [2–4]. Forests serve as essential sinks for carbon, storing approximately 202–275 PgC [5]. These two account for 82% of the global aboveground biomass carbon (ABC) stores if combined with savanna ecosystems (including woody savanna) [5]. Consequently, land cover changes in these areas are regarded as a major source of emission of greenhouse gases [6,7]. These figures indicate the importance of the monitoring and the related aboveground biomass (AGB) in these areas. The spatial assessment of the amount and related dynamics of woody AGB is crucial to project future developments and assess past changes, not only on the local or national level, but also the global level.

Remote sensing techniques have been applied for decades to foster sustainable forest monitoring. Since the 1970's, this was predominantly accomplished by using optical remote sensing systems due to their more extensive archive, accessibility, as well as straightforward interpretation as compared to microwave data [4,8,9]. Thus, forest cover monitoring using optical data from coarse to fine spatial resolution was conducted on the regional to global scale [10–13]. Nevertheless, the impact of persistent cloud cover/haze over many forested areas, leading to gaps in time series analysis, is a major limitation of optical remote sensing data. This problem can be remediated by the joint use of SAR and optical data [10]. In this context, it could be shown that using both, C- and L-Band SAR together with optical data is capable of significantly improving the forest monitoring results [14–16] and minimizing issues of data continuity [17].

With the start of Sentinel-1 and -2, which are acquiring data in the microwave and optical range of the electromagnetic spectrum, data in high geometrical and temporal resolution became freely available. Recent studies demonstrated the potential of Sentinel-2 data for the distinction of land cover classes and forest types with high accuracy when applying single time steps or multi-date information [18–20]. Similarly, various studies analyzed the suitability of C-Band radar data from Sentinel-1 to investigate land cover [15], forest extent [21], forest change [22], deforestation [23], and woody cover [24]. The fusion of both data sources was reported to increase the classification quality in numerous applications, such as crop [25], forest [26,27], and primary vegetation mapping [28]. The existence of more data sources and fast growing data archives naturally led to an increasing demand for machine-learning approaches that are capable of dealing with high-dimensional multi-source data for forest structure related applications [29,30]. A large number of studies utilized such algorithms in the past to map various land cover related metrics while using multi- and hyper-spectral [31,32] as well as radar data [33]. Similarly, these techniques were used to derive forest/land cover changes [6,34–36] and associated parameters, such as tree species [18,37,38] on different scales, woody cover assessments [39,40], or forest habitats [41]. Multiple studies investigated the potential of Sentinel-1 C-Band SAR data to improve the classification accuracies of optically based approaches of estimating tree cover [42] and characterizing forest ecosystems [17,43]. Further, Sentinel-1 proved to perform well in land cover mapping in heterogeneous landscapes, such as the South African savanna [44].

This study aims at deriving forest cover structures in the study sites in Germany and South Africa while using optical (Sentinel-2) and SAR (Sentinel-1) data in independent and joint approaches while using an innovative CV procedure that takes spatial autocorrelation of the data into consideration.

## 2. Materials

### 2.1. Study Sites

The first study site is located in the federal state of Thuringia and is visualized in Figure 1. Located in the center of Germany, it has an area of approximately 16,200 km². Land cover predominantly consists of homogeneous coniferous and deciduous forests, as well as agricultural land. The area is characterized by regional climates that can be described as temperate oceanic, exhibiting warm

summers with dry periods and cold winters. While the northern part of Thuringia, the 'Thuringian Basin', is one of the driest areas of Germany, rainfall accumulates up to 1500 mm per year in the Thuringian Forest, a mountain range located in the south-east of Thuringia. Precipitation is evenly distributed throughout the year [45].

The second study region, visualized in Figure 2, comprises an area ranging from forest plantations in the Mpumalanga province to the southern parts of Kruger National Park (KNP), covering an area of approximately 19,800 km$^2$. In contrast to the Thuringian study site, this site is highly heterogeneous with regard to land cover, the amount of intra- and interannual rainfall, as well as climatic conditions. While KNP is predominantly characterized by patchy patterns of loose aggregations of vegetation and large portions being covered by bare soil (vegetation growth is heavily dependent on seasonal effects), the elevated plateau in the west of the study area features dense forest structures. According to this, rainfall varies between 500 mm and more than 1000 mm for KNP and the outer areas, respectively [46].

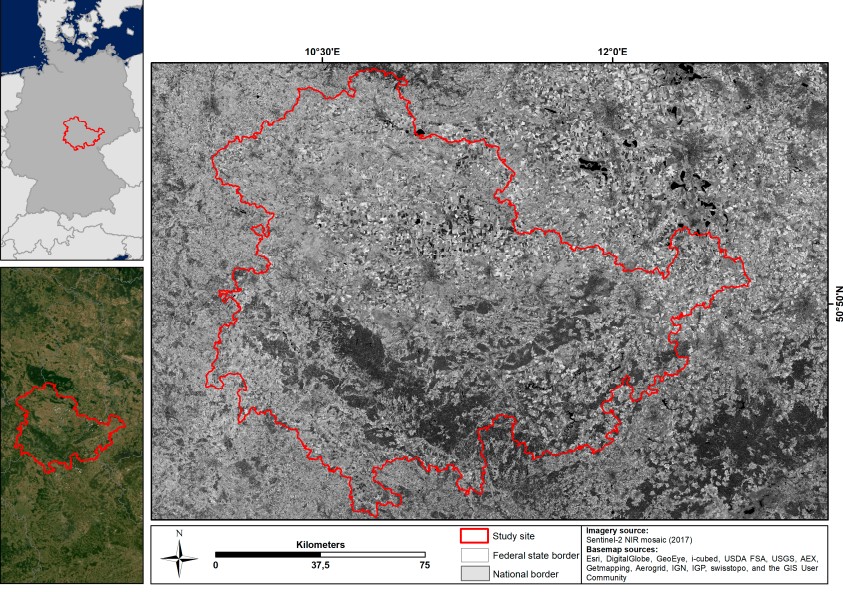

**Figure 1.** Study site in central Germany (Thuringia).

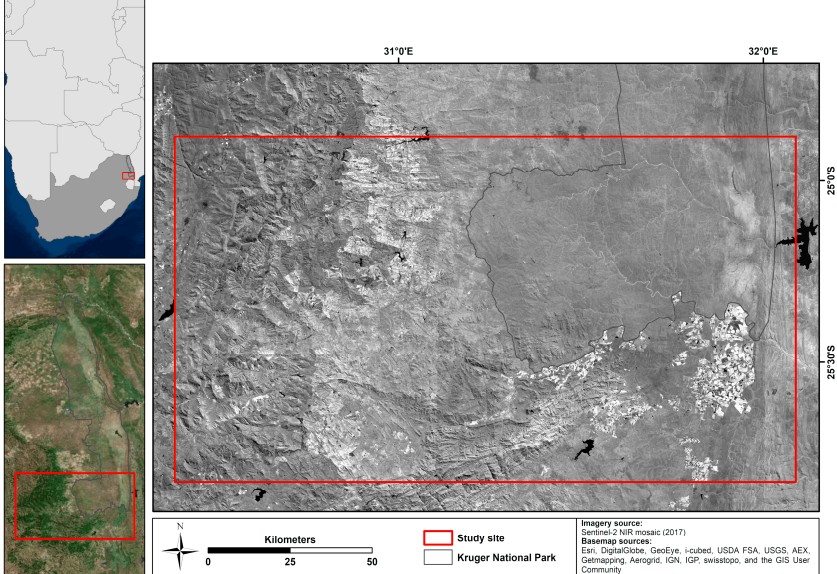

**Figure 2.** Study site in South Africa (southern KNP and forest plantations).

### 2.2. Data

#### 2.2.1. Satellite Data

Data from Sentinel-1 and -2 were acquired for the study sites that are shown in Figures 1 and 2. Both data sources were collected in Level 1 format, which required several preprocessing steps prior to image analysis (see Section 3.1). Dual-polarized (VV + VH) Sentinel-1 backscatter intensity and Sentinel-2 data excluding the 60 m bands (B1, B9 + B10) were used in this study. Table 1 provides an overview of the used predictor variables. Data for both study sites were collected multi-temporally. Sentinel-1A data from 2015 to 2017 were used to calculate multi-temporal metrics for both study sites to capture the temporal dynamics of varying land cover types and, thus, compensating for the noticeable noise effects in the C-Band signal [47]. By using temporal features contrary to the application of individual time steps, it is possible to utilize the information of intra-annual trends within the data. To further analyze the impact of seasonality on the classification, these statistics were separated into winter and summer seasons for the SAR time series. As summer period, we selected the months June to September for Thuringia and November to March for the South African study site (predictor suffix = 'sum'). Winter was defined as the duration from December to March and May to September for Thuringia and South Africa, respectively (predictor suffix = 'win'). Only Sentinel-2A scenes with less than 5% cloud coverage were considered for a mosaic for each study site of the dry season 2016. In total, 127 Sentinel-1A, as well as three Sentinel-2A scenes, and 92 Sentinel-1A as well as four Sentinel-2A scenes, were collected for the Thuringia and the South African study site, respectively.

**Table 1.** List of predictor variables from Sentinel-1 and Sentinel-2.

| Sentinel-1 (per Polarization & Season) | Sentinel-2 |
|---|---|
| minimum | B2 |
| maximum | B3 |
| midhinge | B4 |
| standard deviation | B5 |
| range (95th, 5th percentile) | B6 |
| 5th percentile | B7 |
| 25th percentile | B8 |
| 75th percentile | B8A |
| 95th percentile | B11 |
| | B12 |
| | LAI |
| | FAPAR |

#### 2.2.2. Reference Data

Data provided by the State Office for Surveying and Geoinformation of Thuringia served as reference information for the extent of forests in Thuringia. The data originate from a digital land cover model (DLM) and comprise detailed information regarding the extent of forests as well as tree species on a Sentinel sub-pixel level (<10 m). Located south-east of the city of Jena, the 'Roda' reference site (visualized in Figure 3a), which was selected as training and validation subset for the classification algorithm, extends over an area of 261 km$^2$, of which approximately 50% is covered by coniferous (85%) and deciduous (15%) forest. The forest patch size in the Thuringian training and validation site varies between 0.1 ha and 4173 ha with a median of 1.4 ha.

As in any other protected landscape, freely available in situ data are scarce and difficult to acquire. Therefore, for the South African study site, two sets of reference data were combined to characterize the savanna ecosystem of KNP, as well as dense forest plantations west/north-west of the southern KNP (indicated in Figure 3b). Firstly, forest compartment data from York Timbers forestry company were used to define training areas for homogenous forests, as they can also be found within the German study site. For this reference data set, an age and NDVI threshold was applied to filter compartments

that were covered by mature forests, and that were not logged during the Sentinel-2 acquisition dates and were also not altered throughout the Sentinel-1 time series period. Secondly, vegetation height metrics that were based on airborne Light Detection and Ranging (LIDAR) measurements were obtained from a canopy height model (CHM), which was made publicly available by Smit et al. [48]. Acquisition dates of this data set were April/May 2010, 2012 and 2014, respectively. For this study, exclusively, the vegetation structure from the last mission date was used to minimize the time gap to the Sentinel-1 acquisitions. The available LIDAR coverage is located on the southern boundary of KNP near Malelane Gate and it comprises an area of 1.8 km by 26 km with a spatial resolution of 2 m. In a next step, the vegetation height was limited to a value of more than 5 m according to FAO's forest definitions [49]. Additionally, we followed FAO's forest cover definition with a tree canopy cover of more than 10 percent and an area of more than 0.5 ha. The size of forest patches in the South African varies significantly from the distribution that can be found in the German study site. While forest patches with an area less than 0.01 ha could be found in the KNP, the forest plantations exhibited patch sizes of up to 9350 ha. The median value for the whole study site is less than 0.1 ha. For both reference sites, we extracted 500,000 data points in a stratified sampling approach to be used for hyperparameter tuning, training, and CV, which were carried out after a spatial partitioning.

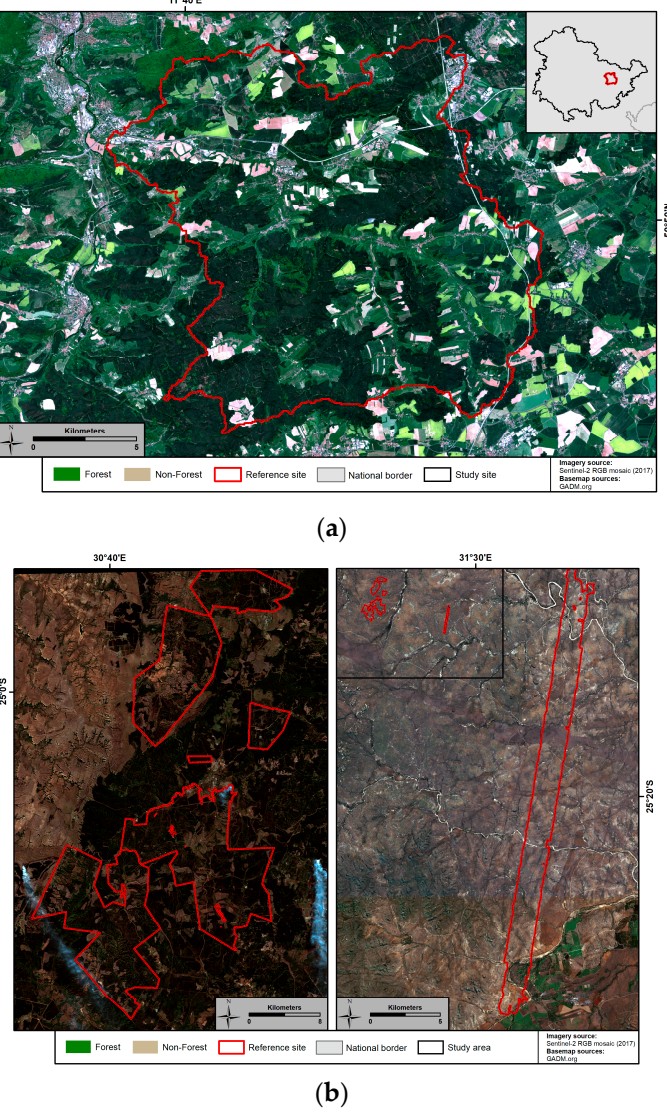

(**a**)

(**b**)

**Figure 3.** Reference data sets for both study sites. (**a**) Thuringia and (**b**) South Africa.



## 3. Methods

The presented methodology comprises the preprocessing of microwave and optical Sentinel-2 data, acquisition of suitable reference information to characterize strongly different ecosystems as well as the introduction of the concept of spatial autocorrelation to perform tuning and validation procedures while taking spatial dependences in remotely sensed data sets into account. Based on this, forest extents were estimated for Sentinel-1, Sentinel-2, and a fused product, including both sensors. Figure 4 provides the overall workflow.

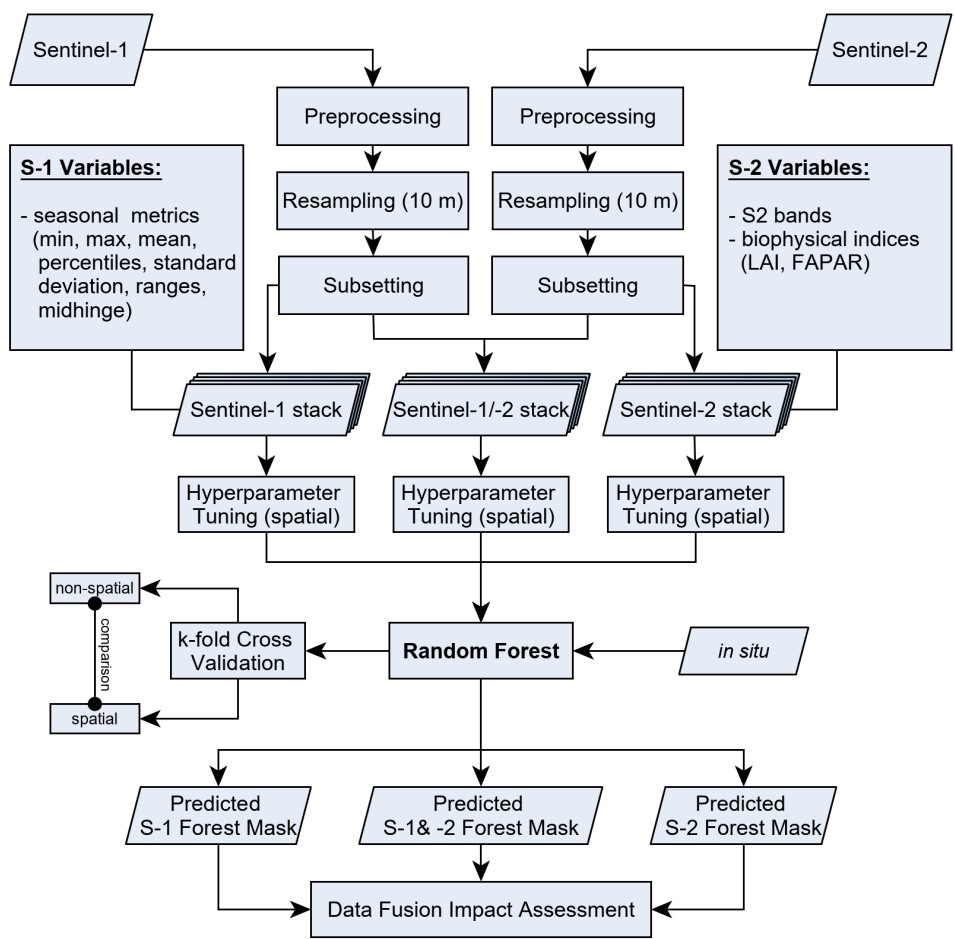

**Figure 4.** Workflow of the forest cover derivation using Sentinel-1 and Sentinel-2.

### 3.1. Preprocessing

Data from both satellites were gathered from ESA's Copernicus Open Access Hub archive. Preprocessing of Sentinel-1 data included multi-looking, geocoding, radiometric calibration, as well as topographic normalization, and it was solely carried out using Gamma routines. Dual-polarized multi-temporal microwave data sets were created at 10 m spatial resolution. Next, Sentinel-2 images were atmospherically, cirrus, and terrain corrected using the Sen2Cor algorithm [50] and Shuttle Radar Topography Mission (SRTM) with a spatial resolution of 1 arc-second ($\approx$30 m). During atmospheric correction, look-up-tables (LUT) for parameters, such as aerosol type and mid-latitude, were automatically chosen based on metadata information. For analysis, Sentinel-2 bands with a resolution of 60 m (channels 1, 9 & 10) were excluded, as their main purpose can be seen within the preprocessing to detect cloud coverage. After preprocessing, multi-temporal Sentinel-1 metrics and biophysical Sentinel-2 indices were calculated and grouped into seasons while using SNAP and R. The latter included the vegetation parameters LAI and FAPAR, which were calculated using the biophysical processor implemented in ESA's SNAP v6.0.0 [51]. As mentioned earlier, the data was

split into seasons, with April and October serving as transition months. This split allowed for a more detailed analysis of the importance of predictor variables with respect to seasonality and the related environmental conditions. Finally, predictor variables from both sensors were stacked and subsetted to the respective study areas in R.

*3.2. Forest Cover Derivation Using Random Forest*

3.2.1. Classifier Algorithm Description and Parameter Tuning

For this study, the decision tree classifier random forest (RF) was utilized while using the ranger implementation within the mlr package [52] of the statistical software R [53]. This non-linear, hierarchical ensemble classifier predicts class memberships that are based on the concept of recursive partitioning to create increasingly homogeneous subsets to retrieve a branched network of data splits [54,55]. Out of all possible splits, the predictor variables are selected (randomly by *mtry*) that minimize the Gini impurity (also referred to as 'splitrule'). Throughout the process, the decision tree is produced independently while being controlled by two main parameters, *mtry* and *ntree*. While *mtry* describes the number of predictor variables that are used to split each node, *ntree* defines the number of trees, which are generally characterized by high variance at relatively low bias [56]. The tuning of these two hyperparameters is crucial, as they control not only the accuracy, but also the computation time of the process [57].

By default, most RF implementations define the square root of the number of predictor variables (limited by number of predictor variables) as the *mtry* value and 500 as the sufficient number of trees (*ntree*). Few studies have investigated the impact of *mtry* towards computation time and the resulting model performance [58,59].

As a first step, the hyperparameters that control the performance of the model needed to be adjusted to retrieve the desired results while maintaining a certain computational effort. We used a repeated CV while using 25 repetitions with five folds to ensure the derivation of stable results, leading to consistent accuracies. Further, the procedure was conducted, including the correction of biases possibly introduced by spatial autocorrelation. For the previously mentioned hyperparameters, *mtry* and *ntree* separate feature spaces, retrieved from literature review, were defined, in which best parameter sets were then estimated [38,59]. Tuning of the variables is crucial for allowing the user the investigation of best performing parameter setups and their related computational costs. Feature spaces were limited *a priori* to keep the amount of processing within bounds that meet existent computation resources. Nevertheless, it should be noted that the hyperparameter tuning, theoretically, should optimally not be restricted to fixed value ranges, since this does require expert knowledge of the process as well as the applied training data. While *ntree* was tuned between 50 and 750, the limits of *mtry* were set between 1 and 5. According to Belgiu and Dragut [29], numerous studies revealed that 500 trees are leading to stable accuracies, which implies that the upper boundary is likely to be found in this value range. Additionally, it was found that the number of trees did not tend to be very sensitive to the prediction outcome when applying high-dimensional Sentinel-1/-2 data in combination to derive forest cover [27]. The *mtry* parameter was limited to only a narrow portion of the value domain, which represents the number of predictor variables within the data set. By default, the RF algorithm sets the *mtry* parameter to the square root of the number of input variables [30]. However, studies have shown that the optimal *mtry* value can be found below this value [60]. Therefore, we set the upper boundary of the search space for this hyperparameter lower than the proposed square root of the number of predictors.

3.2.2. Training and Prediction

Once optimal sets of hyperparameters have been defined for both study sites and the individual sensor setups ((i) Sentinel-1 (S-1), (ii) Sentinel-2 (S-2), (iii) Sentinel-1 + Sentinel-2 (S-12)), these were used to perform the training of the model while using the training data for the study site subsets. The trained

models were then transferred to the respective study regions described earlier. The derived tuning parameters were also applied for tree cover estimation over the complete study site while assuming that the training data was sufficiently representing the regional landscape diversity. The results were further processed while using a sieving algorithm to meet the selected forest definition, which limits the tree aggregations to a minimum size of 0.5 ha and ten percent tree canopy coverage [49].

### 3.2.3. Importance of Predictor Variables

The analysis of the variable importance was carried out based on all datasets to identify which variables were found to be most useful for the distinction between forest and non-forest in each study site. Several filter methods were tested to assess each predictor variable's value for the classification, in order to find most distinctive input variables for the classification. The gain ratio (GR), which was selected to evaluate variable importance, is an entropy-based algorithm that identifies the weights of discrete attributes based on their correlation with a continuous target attribute. GR was utilized to visualize the contribution of each predictor variable and it represents an extension of information gain G, which calculates the average entropy of a single predictor entity A when a set of observations S is split into subsets of $S_i$. The entropy for data sets with C classes is calculated, as follows:

$$E = \sum_{i}^{C} p_i \ \log_2 p_i, \tag{1}$$

where $p_i$ is defined as the probability of the random selection of an element of class i. Consequently, the resulting information exhibits the decrease in entropy of each attribute [61]. The information gain G is then calculated, as given below:

$$G(S, A) = E(S) - E(S, A) \tag{2}$$

G is calculated for every attribute A and the sum of the entropy E with regard to the original set S is then compared to every subset. The predictor entity that maximizes the difference most is selected upon others, being defined as the variable importance within the predictor variable set. However, information gain G is biased whenever the features are branching more complex, as it does not take into account the size of branches as well as their quantity. By using an extension of the previously explained algorithm, the intrinsic information I of each split can be included [61]. The equation of I is given as:

$$I(S, \ A) = -\sum_{i} \frac{|S_i|}{|S|} \log\left(\frac{|S_i|}{|S|}\right) \tag{3}$$

The gain ratio GR then equals:

$$GR(S, \ A) = \frac{G(S, \ A)}{I(S, A)} \tag{4}$$

The results of the calculation of the variable importance are displayed separately for both study sites, respectively, and split up into each specific sensor setup in Section 4.2. Only the seven most important variables were considered and displayed in a combined plot to limit the number of selected predictors.

### 3.2.4. Cross-Validation and Statistical Comparison

A repeated CV was performed using approaches similar to the previously explained tuning algorithm to assess not only the expected classification accuracy, but also to account for spatial dependence in the validation. Within *k*-fold CV the reference information is spatially subdivided into *k* equally sized partitions of which a single subsample is used for testing and the remaining *k* − 1 portions

are used as training information. This process was then repeated *k* times to make use of all available training data and estimated as an average to estimate a single prediction. Validation in this study was accomplished while using a five-fold approach using 25 repetitions. Following, the average error among all *k* runs is computed, adding up to 125 RF trials for each sensor setup and study site. A spatial partitioning of the reference data was carried out to account for the existence of spatial autocorrelation, which is omnipresent in remotely sensed data. The term 'spatial portioning' implies the creation of spatially disjoint training and testing partitions while using a *k*-means clustering approach [62]. Typically, CV approaches are performed using random or stratified sampling methods [34,63,64]. However, these do not take the spatial dependence of observations in spatial data sets into account and therefore ignore the bias introduced by non-spatial sampling [65,66]. Thus, commonly adapted CV lacks in consistently generating training and test folds that are independent from each other [64].

A comparison with other FNF products was carried out to identify the quality of our forest classifications. Thus, our results could be compared to completely independent data sets. For this, 10,000 randomly stratified samples were extracted and then used to compute the Jaccard Similarity coefficient (J) between each other and with respect to the reference data sets for each study site [67]. J is commonly used in data science as a tool to measure similarity/dissimilarity in vectors containing binary values and it was found to provide reliable estimates while preventing overoptimistic results [68]. The Jaccard coefficient ranges between 0 and 1, with 0 representing the lowest possible similarity between quantities. This coefficient measures the similarity between two sets of data by dividing the size of the intersection between data set A and B by the size of the union of both sets.

$$J_{A,B} = \frac{|A \cap B|}{|A| + |B| - |A \cap B|} \tag{5}$$

For comparison, three different common FNF products with similar spatial resolution were chosen. Table 2 provides an overview of these data sets.

**Table 2.** Data sets used for similarity analysis.

| | Author | Pixel Size | Period | Data | Study Region |
|---|---|---|---|---|---|
| Landsat FNF | [10] | 30 m | 2000–2018 | Landsat | both |
| ALOS FNF | [69] | 25 m | 2017 | PALSAR-2 | both |
| Copernicus HRL * | [70] | 20 m | 2015 | Sentinel-2, Landsat, SPOT-5, ResourceSat-2 | Thu |
| CCI ** | [71] | 20 m | 2015–2016 | Sentinel-2 | SA |

Notes: * Copernicus High Resolution Layer (HRL)—Forest Type. ** ESA Climate Change Initiative (CCI) S2 Prototype Land Cover Map of Africa.

## 4. Results

### 4.1. Forest Cover Derivation

The final classification of forest cover was carried out using the hyperparameters *ntree* = 500 and *mtry* = 1, which were found to produce reliable results with high accuracy while maintaining a certain computational effort. Table 3, as well as Figure 5, display the accuracies for the CV of the RF based classifications of forested areas in both study sites using Sentinel-1, Sentinel-2 and a fused data product. The overall accuracy was calculated using the median value of all runs of the spatial CV, since this provides a more realistic representation of the actual classification quality, as outliers tended to appear especially in the South African study site.

**Table 3.** Averaged results (median) for each sensor setup from all CV repetitions.

| | Sentinel-1 | Sentinel-2 | Sentinel-1/-2 | |
|---|---|---|---|---|
| **South Africa** | 84.3 | 90.4 | 92.3 | 90.9 |
| **Thuringia** | 90.6 | 93.3 | 93.7 | 93.2 |
| | 87.5 | 91.9 | 93 | Ø |

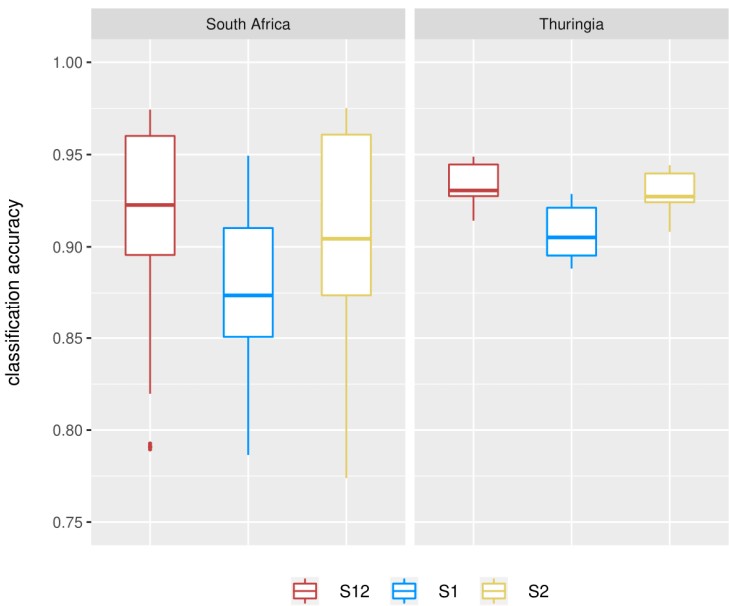

**Figure 5.** Results of CV for both study sites comparing all sensor setups.

Classification accuracies were found to be consistently higher in the German (93.2%) as compared to the South African study site (90.9%) when being averaged over all available sensor setups. Considerably strong variations between these configurations could be observed in both study sites. As visualized in Figure 5, the standard deviation, which was calculated over all CV runs, was significantly larger in the South African study site (0.33), while the results in Thuringia did not differentiate between all CV runs as much (0.012). The difference in the variance visible in the boxplot range can be attributed to the existence of more sufficient training data in the German study site, as well as the appearance of forests in this area, which is much more homogeneous when compared to the South African study site, making it easier for the algorithm to reliably detect forest based sites. It should be noted that the plot is cropped to a lower boundary of 0.75 for visualization reasons as very few values of the Sentinel-1 based classification in South Africa ranged around 0.2. In both study sites, Sentinel-2 tended to outperform Sentinel-1 in the single-sensor approach leading to higher classification accuracies. While the fusion of Sentinel-1 and Sentinel-2 led to the highest classification accuracy in the German site, the potentially lower overall classification accuracy in South Africa also impacted the fused approach.

Figure 6 displays the results of the classifications using all available combinations of sensor setups for the South African study site. Here, Sentinel-1 predicted a greater amount of forest extent in parts of the KNP as well as the area just outside the park (brown) when compared to setups utilizing optical data. It can also be seen that Sentinel-2 only classified few forests/tree aggregations in the reserve area while capturing forest plantations in the west of the KNP very well (dark green). The optical sensor also misclassified agricultural fields that were located south of the Kruger National Park. This was not visible in the multi-temporal SAR classification. Combining both classifications, this was found to positively impact the optical classification. The forest prediction using Sentinel-1 and Sentinel-2 showed a good fit with homogenous forests in the western part of the study area and an underestimation of forest in the Kruger National Park.

In the Thuringian study site, the algorithm achieved higher overall accuracies compared to the South African study area. Results are visualized in Figure 7. Here, it was found that the optical, microwave and the fused FNF classification produced quite similar results, which was also reflected in the accuracies (see Table 3). While the SAR based forest cover map (brown) showed a larger number of noisy and, thus, misclassified pixels, gaps in the optical data caused by cloud cover could be filled by applying data fusion prior to the classification.

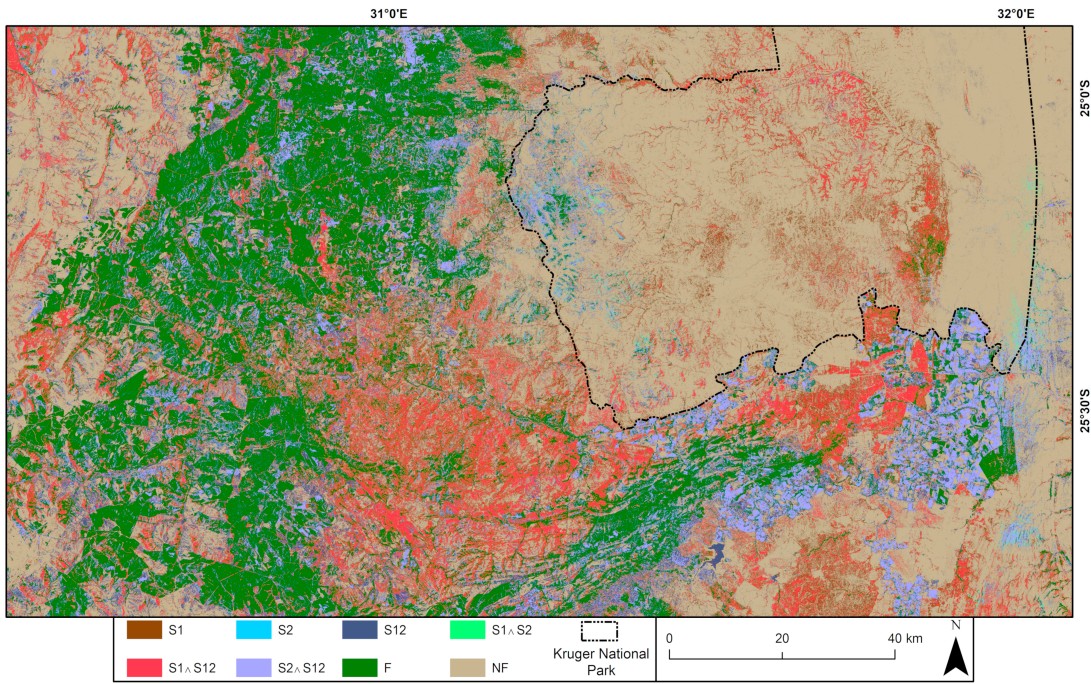

**Figure 6.** FNF map of the South African study site showing pixels being classified by Sentinel-1, Sentinel-2, a fused product (S12) and combinations of these setups; 'F' represents pixels classified as 'forest' by all sensor setups; 'NF' represents the pixels classified as 'non-forest' by all sensor setups.

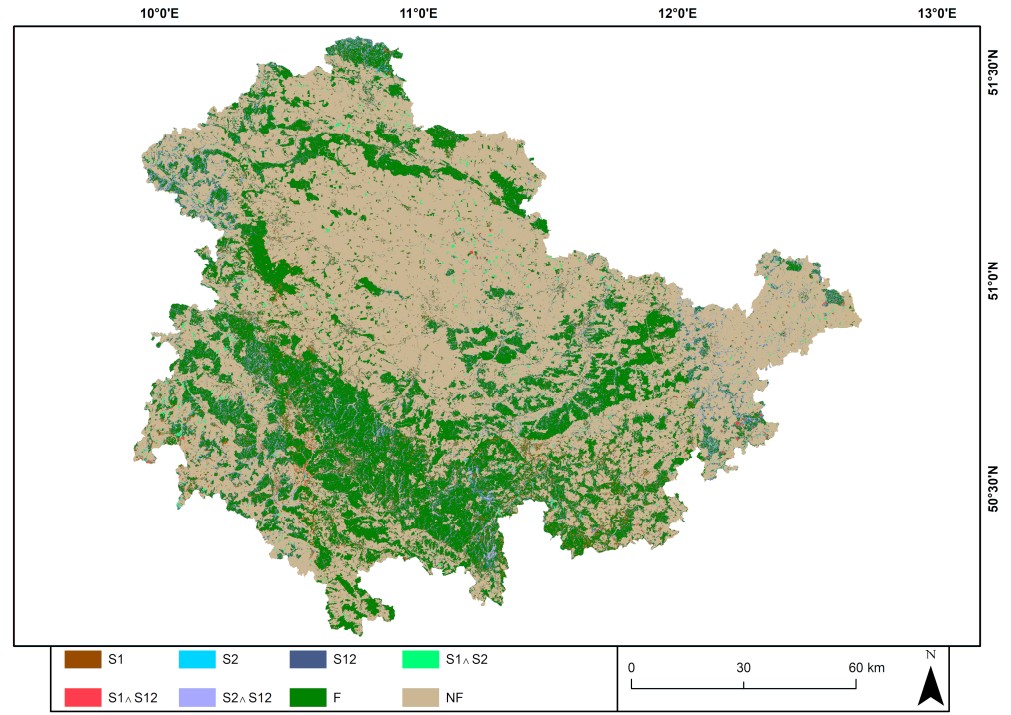

**Figure 7.** FNF map of Thuringia showing pixels being classified by Sentinel-1, Sentinel-2, a fused product (S12) and combinations of these setups; 'F' represents pixels classified as 'forest' by all sensor setups; 'NF' represents the pixels classified as 'non-forest' by all sensor setups.

The discrepancy between varying single sensor setups (microwave vs. optical) was found to be much stronger in the more heterogeneous landscape of the South African study site comprising savanna and forest ecosystems. Here, a sub-classification was performed dividing the study site

into two parts, a) the forested areas spreading over from Mpumalanga to the province of Limpopo and b) the savannas of the southern KNP. The results suggested that both sensors produced reliable classifications in the western part of the study site while the savanna part and its vegetation was strongly underestimated by the Sentinel-2 predictors. In contrast, Sentinel-1 C-Band data were able to capture greater portions of tree aggregations in this part of the site. However, the Sentinel-1 setup might led to misclassifications in open savanna, due to limitations of the savanna LIDAR training data set to represent especially lower vegetation (<2 m) and the strong impact of soil moisture, as well as surface roughness. Forest sites with denser canopies were found to be underestimated by the RF classifier when applying SAR data only in both study sites. Here, closed tree canopies force a limited penetration of the C-Band signal. This corresponds to findings of other studies working in this area [39]. Nevertheless, Sentinel-2 outperformed Sentinel-1 only classifications in all study sites in terms of accuracy. This can partly be attributed to the internal variable selection of the RF algorithm that was based on the underlying training data. Consequently, the distinction of FNF in both study sites relied stronger on optical predictor variables, despite the information content stored in the SAR time series.

## 4.2. Analysis of Variable Importance in Varying Sensor Setups

### 4.2.1. Thuringia

Variable importance in both study areas was calculated based on the equations that are given in Section 3.2.3. As visualized in Figure 8, variables that represent the lower range of the SAR time series, such as the minimum or the 25th percentile of the summer season, were found to be the most distinctive predictors for the Sentinel-1 time series in this study site. Here, GR does not vary significantly between the seven most important variables, leading to comparable contributions to the classification result. As the minimum backscatter values throughout the summer months tend to occur towards the end of the growing season for C-Band similar to L-Band SAR data when the signal received from ground surfaces is increased, a better distinction can be found during this period of the year [72]. It was also found that polarization and orientation of the signal being received and emitted do strongly influence the detection of plant structure. Predictors with VH polarization showed to be the most relevant SAR variables for the classification in the German study site. This corresponds to findings of Olesk [22], who found that cross-polarized SAR data from Sentinel-1 is more suitable in most cases when detecting forests as compared to like-polarized microwave radiation.

The analysis of variable importance of the Sentinel-2 classification revealed that the biophysical parameters FAPAR and LAI, which are directly relatable to photosynthetic primary production and activity, exhibited the best differentiation for Thuringian temperate forests [73]. The latter proved to be the most important predictor for forest cover in this study site, which is due to the homogenous tree canopies that can be found in this study site with GR of above 0.3. SWIR (B11, B12), RGB (B2, B3, B4), as well as the shortest red edge band (B5) were also of great importance for the classification performance in the optical setup. Recent studies confirm our findings, which consider the SWIR bands to be among the most important Sentinel-2 channels for vegetation cover mapping [18,74]. The near-infrared (NIR) channels B8 and B8A were found to be only partly capable for a FNF distinction when being compared to the remaining channels of Sentinel-2, which does not correspond with other studies, as these wavelengths tend to be most representative for photosynthetic active vegetation [75].

Analysis of the combined predictor variables from Sentinel-1 and -2 showed that the GR of optical predictors clearly outperformed Sentinel-1 features in terms of variable importance, which was also visible in the final classification. Besides two SAR features (min_VH_sum and p25_VH_sum), optical variables or their derivatives were found to be the most dominant in the classification process and hence the prediction. As Sentinel-2 was favored over the SAR variables in the classification process, this effect could also be seen in the variable importance of the S12 sensor setup (Sentinel-1 and Sentinel-2). Here, no significant differences between the Sentinel-2 variables (SWIR and RGB) with the highest

importance were observed, while LAI and FAPAR remained as the most important variables in the optical sensor setup. In general, the variables that were most distinctive in the single-sensor approach of the optical data were also found to be the most important ones for the fused approach.

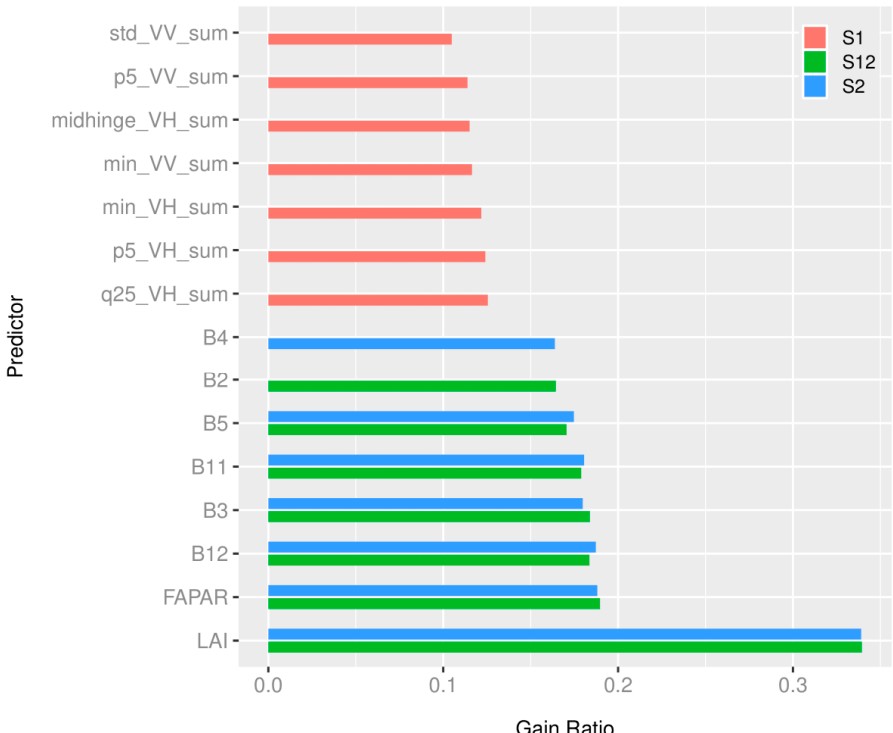

**Figure 8.** Variable importance in the Thuringian study site (seven best predictors).

### 4.2.2. South Africa

Variable importance was separately analyzed for the KNP and the remaining areas of this study area to account for the substantial differences in the appearance of these landscapes due to the great heterogeneity of the South African study site. Figures 9 and 10 display the feature importance for each of the three applied sensor setups in the South African sub study sites.

Similar to the Thuringian study site, the Sentinel-1 features exhibited higher GR values when they were derived from VH-polarized variables in both sub sites. As VH-polarized C-Band data is more sensitive to volume scattering and, therefore, able to monitor vegetation structure it was more important during the classification when compared to co-polarized data. Due to the interaction of cross-polarized predictor variables with shrubs covering large portions of the Kruger National Park, these were found to be more important in the model predictions within the protected savanna ecosystem as compared to their optical counterparts. Between the multi-temporal SAR metrics, differences in GR were relatively small, while variables representing the dry season (winter) ranked higher. This might be due to differences in the vegetation status during the dry season between tree aggregations and their surroundings, thus making it more feasible for the algorithm to detect forests with higher accuracy. Several studies confirm the findings stating that the 'leaf-off' season should be favored for monitoring vegetation structure in savanna ecosystems with SAR information [76–78]. This can be attributed to the increased transparency of deciduous tree canopies and the associated increased penetration depth of the C-Band signal, as well as the decrease of the impact of soil moisture change in this period [75,79]. Consequently, Sentinel-1 was also seen to be more important in the fused approach while using both sensors in combination in the sub site of the open savanna. These results indicate the potential of multi-temporal microwave remote sensing instruments to improve vegetation monitoring in savanna ecosystems.

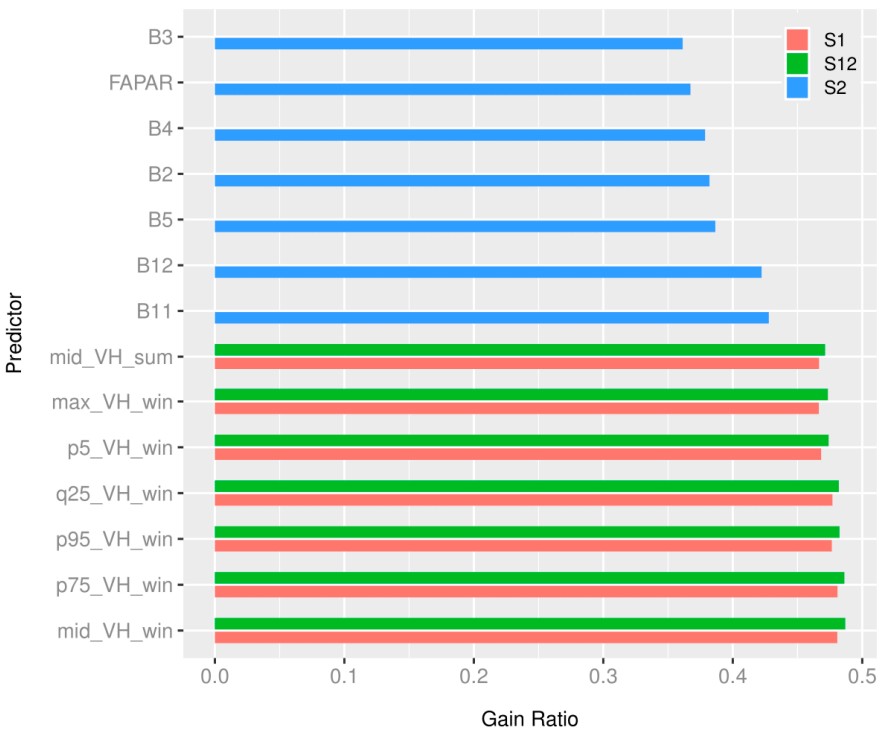

**Figure 9.** Variable importance within the KNP (seven best predictors).

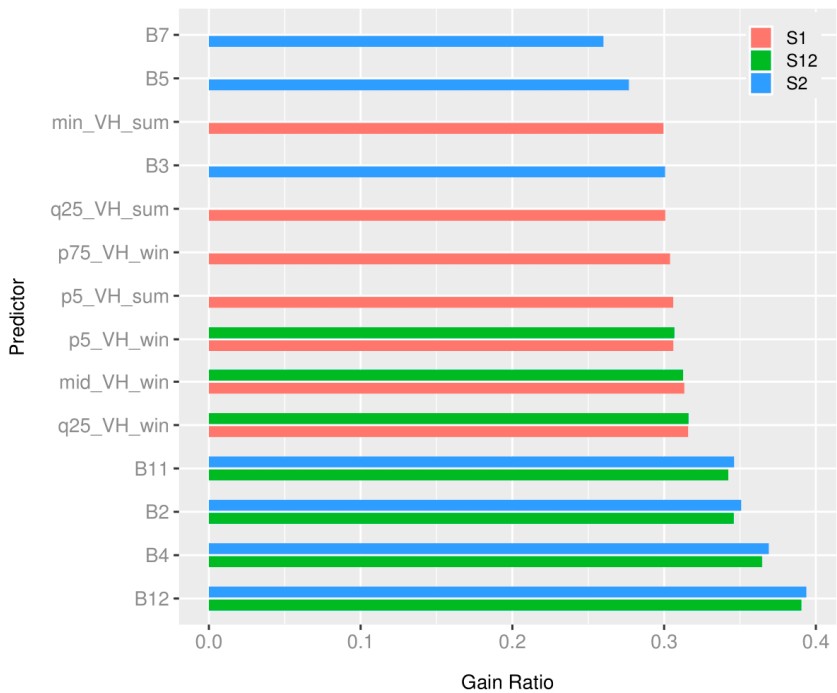

**Figure 10.** Variable importance outside of the KNP (seven best predictors).

Outside the KNP (west), variable importance shifted strongly towards the use of Sentinel-2 predictors by the RF algorithm, as visualized in Figure 10. Similar to the Thuringian study site, the results indicated that the SWIR bands 11 and 12, as well as the channels in the visible bandwidth, are the most important bands for the optical and the fused classification. In contrast, the biophysical parameters were not as distinctive for the classification, as it was observed in Thuringia. This could possibly be explained with the highly heterogeneous composition of land use and land cover in this area. While, dense forest plantations in the west can be represented well by using these indices that

can be directly related to vegetation structure, extensive transition regions towards the savanna flora are often characterized by scarcity in terms of plant size and covered area. Information from NIR wavelength were found to be of small importance for the distinction between forest and non-forested areas, which is consistent with the analysis in temperate German forests that was conducted in this and other studies [18]. As this effect could be observed in both study sites, this suggests that the near-infrared channels of Sentinel-2 are only of limited significance for the derivation process of tree cover or tree species mapping when SWIR channels are being implemented in the classification.

Sentinel-1 predictors representing the lower ranges of the SAR time series in VH polarization were also found to be superior to the co-polarized data in terms of variable importance. In contrast, to the protected park area in the east of the South African study site, dry and wet season were found to equally contribute to the model performance. This might also be related to the stability in the forest plantations and their surroundings originating from the weaker seasonal changes of vegetation in this area.

### 4.3. Comparison with Existing FNF Products

In comparison with the chosen reference products, the FNF classifications using Sentinel-1 and Sentinel-2 produced higher J values (Ø = 0.87 vs. Ø = 0.77) in both of the study areas. As given in Figure 11, the similarity of a given set of samples of reference with the data sets reveal a distinct relationship between the composition of the study sites and used sensors. While J of classifications based on mostly optical data (S-2, CCI, Landsat) exhibited generally higher values in the Thuringian study site than SAR based (Ø = 0.9 vs Ø = 0.8), this trend was found to be reversed in the South African study site (Ø = 0.68 vs Ø = 0.88), which exhibits a much higher degree of homogeneity. This indicates the ability of multi-temporal SAR data to detect vegetation in savanna ecosystems adequately representing the intra-annual variability of this complex ecosystem better and, thus, showing that optical data might not be sufficient in this area, at least when few time steps are utilized.

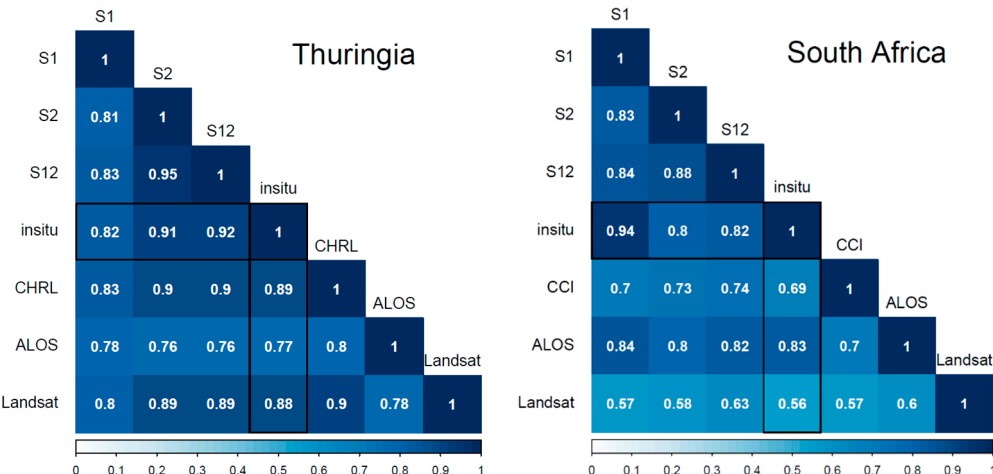

**Figure 11.** J for classifications of all sensor setups (Sentinel-1, Sentinel-2, and Sentinel-1/-2) in the Thuringian and South African site compared with reference data and independent FNF products.

### 4.4. Test of Homogeneity between Classification Distributions

To assess whether the results of the classifications vary significantly from each other, we performed McNemar's test to each individual sensor setup. This non-parametric statistical test is suited to compare the performance of machine-learning-based classifications [80]. The results of McNemar's test indicate that, except for the comparison of classification S-2 vs. S-12 in Thuringia, all of the classifications were considered to significantly differ from each other.

## 5. Discussion

This study investigated the use of multi-source and multi-temporal remote sensing data in varying ecosystems. Firstly, the results revealed that the overall accuracy of the machine-learning based classification while using the RF algorithm increased from using single sensor setups as compared to using Sentinel-1 and Sentinel-2 in a joint approach in both study sites. Combining optical and SAR data led to very high classification accuracies, as illustrated in Table 3. Comparing all available sensor setups, the Sentinel-1 classifications provided the lowest accuracy while still performing reasonably (84.3% to 90.6%) in both study sites. While Thuringia is characterized by relatively homogenous forests, the South African study site comprises forest plantations and savanna ecosystems, thus exhibiting a gradient of increasing heterogeneity from East to West, which also led to great differences in the variable importance for the RF model. In both study sites, the SWIR bands 11 and 12 of Sentinel-2 ranged the highest among the most distinctive optical variables, which other studies also confirmed [18,26]. From Sentinel-1, VH polarized variables were selected as the most important variables in the single sensor and fused data set.

The savanna is a highly variable ecosystem in terms of vegetation composition, as reflected in a non-trivial spectral and spatial appearance, as well as phenology, which drastically changes between dry and wet season [81]. Subsequently, classification results of this study site featured the highest variability in CV-runs over all sensor setups, as visualized in Figure 5.

Further, it could be observed that the SAR data was capable of detecting smaller aggregations of trees in the scarce South African savanna than its optical counterpart, while overestimating tree cover in this area. This can be mainly attributed to the impact of surface roughness to which the C-Band radar is very sensitive, especially in sparsely vegetated areas. This effect was also shown in a study that was conducted in savanna ecosystems using Sentinel-1/-2 data [82]. Due to the heterogeneous training data, which may not represent the complete study area adequately, the accuracy does not fully reflect the positive impact of Sentinel-1 on the savanna classification. Here, the quality of the training and validation data is a major key in obtaining satisfactory and reliable classification results [83]. Thus, a separated variable importance analysis for the South African study site proved the significant role that SAR data plays in the detection of vegetation in open savannas. It should be noted that a separation of the South African study site into two internally more homogenous study areas would potentially lead to an increase in classification accuracy and a strong decrease of the root mean square error.

The high-resolution optical Sentinel-2 data proved to be capable of detecting forests with high accuracy in both study sites, especially in areas with homogenous forest sites. It is important to highlight that Sentinel-2 tends to underestimate savanna vegetation during dry season with the exception of larger tree aggregations, which are often located along river streams. Switching from single time steps to deploying the fully available cloud-free time series of Sentinel-2 data might lead to an improvement to further increase classification accuracy in this ecosystem and, thus, improve the ability to take the strong seasonality of South African savannas into account.

This study further analyzed the impact of sensor fusion of Sentinel-1 and Sentinel-2 to improve forest monitoring in highly variable ecosystems. Similarly to findings of forest monitoring related studies while using optical and radar satellites, the addition of Sentinel-1 data did not significantly improve the overall classification accuracy [26,28]. In both studies, differences between the accuracy of a Sentinel-2 only and fused classification (Sentinel-1 and Sentinel-2) ranged between 1 to 3 %. These results are comparable with our findings. By joining optical and SAR data, the dense Sentinel-1 time series could capture vegetation dynamics in open savanna while being prevented from possible overestimation by Sentinel-2 when vegetation is scarce. The results also indicated that sensor fusion increases the classification accuracy in both study sites averaged over multiple runs of CV. The accuracies show that the individual classifications of sensor setups S-2 and S-12 did not differ as much when compared to the S-1 classification in both study sites. A statistical McNemar test was conducted to check the homogeneity of the distribution of the classifications. The results show that all of the distributions differ (given a significance level of $p = 0.05$ and one degree of freedom) significantly from

each other (except for S-2 vs. S-12 in Thuringia), as given in Table 4, with S-1 classifications exhibiting the greatest differences. This indicates the different perception and potential that is provided from the sensors for the varying ecosystems.

**Table 4.** McNemar's test results for varying sensor setups in both study sites. A *p*-value greater than 0.05 defines a difference between two distributions that is not significant and vice versa.

| Product | Thuringia | | South Africa | |
|---|---|---|---|---|
| | $X^2$ | *p*-Value | $X^2$ | *p*-Value |
| S-1 vs. S-2 | 48.3 | <0.05 | 61.6 | <0.05 |
| S-1 vs. S-12 | 66.9 | <0.05 | 57.2 | <0.05 |
| S-2 vs S-12 | 1.4 | 0.23 | 4.8 | <0.05 |

The results of this study demonstrate the ability of machine-learning techniques to produce reliable results from a large number of variables given a relatively low quantity of reference information. Our findings suggest that the RF algorithm favored optical data over multi-temporal SAR data for the detection of forests in the different ecosystems of both study sites. Further, C-Band was found to be a promising data source for the detection of vegetation in dry savanna ecosystems. However, due to the internal variable selection of the RF classifier, this was not acknowledged strong enough within the classification itself. Using in situ data that provides a better representation of the study area might increase the impact of applying a dense Sentinel-1 time series, so that more of the pixels showing strong intra-annual dynamics are being used for training of the model.

## 6. Conclusions

In this study, we investigated the ability of high resolution optical and microwave Sentinel data to derive forest cover in substantially different ecosystems while using machine-learning techniques accounting for the impact of spatial autocorrelation during cross-validation. As study sites, we selected the state of Thuringia in Germany, which is characterized by homogenous dense temperate forests, and an area in South Africa including the southern Kruger National Park, as well as neighboring forest plantations, featuring both homogenous tree aggregations and scarce open savanna vegetation. The results indicated that optical sensors are capable of detecting homogenous tree aggregations with high accuracies while failing at locating large portions of tree cover in open savannas. The addition of multi-temporal microwave information to this data set showed multiple advantages. These are the correction of falsely classified cloud pixels, as well as an improved delineation of small forests in the savanna ecosystem. Thus, our results show that the fusion across wavelengths can lead to classifications with a minimized quantity of misclassifications, while the magnitude is rather small, especially when comparing optical and fused classification. This finding is also reflected in considerably high accuracies of classifications using the joint data sets, which were all cross-validated with multiple repetitions to avoid spatial redundancy and account for outliers. The analysis of variable importance revealed that SWIR and RGB channels range among the most important predictors from Sentinel-2, which corresponds to the findings of other recent publications. The biophysical parameters used in this study (LAI and FAPAR) were found to be useful in detecting forest cover mainly in homogenous temperate environments. As most important multi-temporal Sentinel-1 features, the classifier identified VH-polarized predictors and those representing the lower value range of the time series, such as minimum and the lower percentiles in both study sites. Sentinel-1 variables were not favored strong enough to reveal their full predictive power in some parts of our study sites due to a certain level of 'black box' behavior of the RF algorithm. We further split the South African study site into two parts to reveal their potential impact on the classification, so that the microwave predictors proved their capability to predict tree cover in open savanna ecosystems.

This study demonstrated the beneficial effects of synergistically combining Sentinel-1 and Sentinel-2 to detect forest cover at fine spatial scale. Using CV procedures that account for the existence

of spatial dependence within remote sensing data sets our methodology could potentially contribute to improve reliability of activity data (AD) under REDD+ Measurement, Reporting, and Verification. We also applied the robust RF classifier to highly variable ecosystems to examine the robustness of the approach. Future studies could potentially focus on extending from Sentinel-2 time steps to proper time series information to obtain an even better understanding of intra- and inter-annual alterations in the vegetation status. Furthermore, it would be essential to gain more control over how predictor variables are used in machine-learning approaches and, thus, fully reflect their importance in the individual classifications (single sensor) to increase the beneficial effect on the fused models (multi-sensor).

**Author Contributions:** Conceptualization, K.H., M.U., M.D.M. and C.S.; Data curation, K.H. and M.U.; Formal analysis, K.H., M.U. and M.D.M.; Funding acquisition, C.S.; Investigation, K.H.; Methodology, K.H., P.S. and M.D.M.; Software, K.H. and P.S.; Supervision, M.U., M.D.M. and C.S.; Validation, K.H.; Visualization, K.H. and M.D.M.; Writing—original draft, K.H.; Writing—review & editing, K.H., M.U., P.S., M.D.M. and C.S. All authors have read and agreed to the published version of the manuscript.

**Funding:** This research was funded by the International Max Planck Research School for Global Biogeochemical Cycles (IMPRS-gBGC).

**Acknowledgments:** Co-Authors received funding from the German Federal Ministry of Education and Research (BMBF) in the framework of the Science Partnerships for the Assessment of Complex Earth System Processes (SPACES2) under the grant 01LL1701A to D (South African Land Degradation Monitor (SALDi)) as well as from the European Union (EU) Horizon 2020 Research and Innovation Program under the Grant Agreement No. 640176 (BACI—Towards a Biosphere Atmosphere Change Index).

**Conflicts of Interest:** The authors declare no conflict of interest.

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
