# Peer review of "Predicting Forest Cover in Distinct Ecosystems: The Potential of Multi-Source Sentinel-1 and -2 Data Fusion"

_remotesensing, doi:10.3390/rs12020302_

Round 1

Reviewer 1 Report

The present paper makes an interesting contribution to the evaluation of Sentinel data with machine learning techniques. It is well written, takes up a current research issue, and will certainly contribute to the further spread of the machine learning techniques in the analysis of satellite data.

As usual, any paper, however good, can be improved. The authors could take up the following points as suggestions for improvement:

Sentinel data will play a prominent role in the future Copernicus programme of the EU and thus in the application of remote sensing in IPCC AFOLU reporting (keyword: REDD+). The approach presented here for recording forest area change can improve the derivation of activity data and especially the reliability. Therefore I would add confusion matrices to the text parts on the verification of the results - these are required, for example, in the methodological framework for REDD+ monitoring by FCPF (FCPF, 2016). Similarly, not only changes in percent forest cover but also in forest area - including 95%-confidence intervals -could be presented. This would significantly extend the practical scope of the methods presented. The description lacks information on the spatial patterns of forest areas in the test areas. Minimum, maximum and median of forest patches should be presented – or as an alterative frequency diagrams for patch sizes in the test areas. This would be useful, as the accuracy of the classification clearly depends on the spatial patterns of classes. The statements on significance between the alternatives used (S1, S2, S12) in Figure 5 (Boxplots) and Table 4 (McNemar's test) differ. This difference must be explained.

I am struggling to recommend a minor or major revision. I finally decided to go for a major revision, but I believe that the changes are very easy to make.

Further comments:

Line 39 ff

The decreasing trend of forest cover as presented by FAO refers to the net change of forest area (i.e. change = increase in forest cover by afforestation – decrease by deforestation). The reference to natural forests and plantations does not explain this fact. The sentence “These estimates….extent of 0.5 ha” could be replaced by “These estimates include decrease in forest area by deforestation and increase by afforestation …”

Line 46-47

The sentence does not read well.

Line 54

“forest cover monitoring” instead of “forest monitoring”

forest monitoring includes both in-situ assessments and remote sensing image analysis

Line 83 ff

Add information on spatial patterns of forest patches

Line 147 -148

FAO does not provide “forest height definitions”. 5m are the threshold for tree height to be reaches in maturity.

Line 151

“cross validation (CV)” instead of “CV”

Line 186

Add citation for R “(R Core Team, 2017)”

…and add name and citation for R-package for RF

Line 209 to 210

Give justification why ntree was set to “50 and 750” and mtry to “1 and 5”

Line 237 to 250

Provide equations for calculating E(S), E(SA), and I(S,A)

Line 254

How was the subdivision of the population performed? Randomly? Systematic?

Line 268

For the further understanding of the Jaccard similarity index a sentence like

“The closer the Jaccard coefficient is to 1, the greater the similarity of the quantities. The minimum value of the Jaccard coefficient is 0.” would improve the understanding, especially of the valuable information given in Figure 11.

Line 454 to 460

The results of the McNemar test contradict the findings presented in Figure 5. A reason could be that the McNemar test assumes correlated observations.

Line 461

Just a suggestion – mention the assessment of activity data (AD) under REDD+ Measurement, Reporting and Verification. Your paper contains an extremely valuable approach to improve the reliability of AD measurement.

Helpful references:

FCPF, 2016. Carbon fund methodological framework. In: FCPF (Ed.), https://www.forestcarbonpartnership.org/carbon-fund-methodological-framework.

R Core Team, 2017. R: A language and environment for statistical computing. In, https://www.R-project.org;. R Foundation for Statistical Computing, Vienna, Austria.

Author Response

Thank you very much for your helpful suggestion. Please see the attached document to find my replies.

Reviewer 2 Report

please see my comments in the attached PDF. Thanks.

Author Response

Thank you very much for your helpful suggestions. Please see the attached document to find my replies.

Reviewer 3 Report

The manuscript investigated the forest detection capacity of Sentinel I and II images using random forest, a machine learning, method in a homogeneous and heterogeneous forest in different climatic conditions. There is not any innovation in terms of the methodology adopted and the results obtained, however, a comprehensive analysis was done in a well-organized way. The manuscript is prepared well and written smoothly still, before going for the publications I would love to see the answer in the following issues.

I don’t find which microwave information did you use in this work, is it backscattering intensity or the backscattering coefficient?

Referring to table 2, the accuracy of South Africa is 90.4 and Thuringia is 93.3 while the accuracy obtained after the data fusion (using of sentinel 1 and 2) is 92.3 and 93.7 respectively. I think the results of sentinel 2 and the fusion are the same, this (less than 2%) difference could have been from the sampling bias of the reference dataset. How do you justify the requirement of the optical and SAR data fusion for forest detection? For me, the research does not contribute anything. Is there anything special that really makes a difference while using the data fusion method?

What is the meaning of column four without heading in table 2?

How did you select the S1 and S2 variables? Does there any basis while considering these variables?

The caption of figure 9 seems to be incomplete.

Line 481 and 482 are ambiguous.

Author Response

(The authors gave the same response as above.)

Round 2

Reviewer 1 Report

Congratulations. All comments have been incorporated and the manuscript is now ready to be published.

Author Response

Thank you very much for the helpful suggestions throughout the review process.

Reviewer 2 Report

Thanks you very much for your revision.
There is only one comment as following:

The resolution of Figures 5, 7, 8 , 9 and 10 should be improved. The font of the y-label is suggested to be the black color font.

Author Response

Some of the figures were added with higher resolution. However, the images were imported into the document with at least 300dpi and it seemed like the Word file lowered the resolution significantly. Thanks for this hint.
The font color of the y-Axis has not been changed from grey to black for consistency reasons and it is also readable in black/white print.

Thank you very much for the helpful suggestions throughout the review process.